# Dynamic Resistance Exercise Alters Blood ApoA-I Levels, Inflammatory Markers, and Metabolic Syndrome Markers in Elderly Women

**DOI:** 10.3390/healthcare10101982

**Published:** 2022-10-09

**Authors:** Nayoung Ahn, Kijin Kim

**Affiliations:** Department of Physical Education, Keimyung University, Daegu 42601, Korea

**Keywords:** dynamic-resistance exercise, elderly women, metabolic syndrome, ApoA-I, inflammation, physical fitness, dysfunctional HDL-C, cardiovascular risk, vascular endothelial function, antioxidant function

## Abstract

Combined endurance and dynamic-resistance exercise has important anti-inflammatory effects, altering vascular endothelial function, and helping to prevent and treat aging-related metabolic syndrome (MS). We studied changes in 40 elderly women aged ≥ 65 years (control group (no MS), *n* = 20, mean age: 68.23 ± 2.56 years; MS group, *n* = 19, mean age: 71.42 ± 5.87 years; one left). The exercise program comprised dynamic-resistance training using elastic bands, three times weekly, for six months. We analyzed body composition, blood pressure, physical fitness, and MS-related blood variables including ApoA-I, antioxidant factors, and inflammatory markers. After the program, the MS group showed significant reductions in waist-hip ratio, waist circumference, diastolic blood pressure, blood insulin, and HOMA-IR, and a significant increase in HSP70 (*p* < 0.05). Both groups showed significant increases in ApoA-I levels, ApoA-I/HDL-C ratio, SOD2, IL-4, and IL-5 levels (*p* < 0.05). Active-resistance training-induced changes in ApoA-I were significantly positively correlated with changes in HDL-C and HSP70, and significantly negatively correlated with changes in triglycerides, C-reactive protein, and TNF-α (*p* < 0.05). Active-resistance training qualitatively altered HDL, mostly by altering ApoA-I levels, relieving vascular inflammation, and improving antioxidant function. This provides evidence that dynamic-resistance exercise can improve physical fitness and MS risk factors in elderly women.

## 1. Introduction

Among cardiovascular risk factors, high-density lipoprotein cholesterol (HDL-C) is considered a key positive health indicator. Reduced HDL-C increases the risk of coronary artery disease (CAD) [1,2], while increased HDL-C reduces the risk of CAD and myocardial infarction. While the inverse relationship between HDL-C and cardiovascular disease (CVD) has long been known [3], the qualitative characteristics of HDL-C are more relevant in preventing CVD than its quantitative characteristics [4,5]. These qualitative characteristics, therefore, provide important prognostic factors of CVD [6]. Further, the structure of HDL-C can affect its cardioprotective function [7].

HDL-C has anti-atherogenic [8], anti-inflammatory, anti-apoptotic, and anti-thrombotic effects [9,10,11,12]. In a normal environment, HDL can prevent oxidation of low-density lipoprotein (LDL), whereas, in pathological states such as oxidative stress, inflammation, or diabetes, it exhibits dysfunctional HDL-C structure, with reduced anti-arteriosclerosis and anti-inflammatory functions [6]. Dysfunctional HDL-C is associated with reduced apolipoprotein A-1 (ApoA-I) and increased serum amyloid A (SAA). Changes in levels of dysfunctional HDL-C can be quantitatively determined by analyzing ApoA-I and SAA levels, by separating the small, high-density subtype HDL3 [13,14]. HDL undergoes structural and compositional changes that alter its anti-arteriosclerosis and anti-inflammatory effects. While it is important to examine how HDL levels, substructure, and composition alter metabolic syndrome (MS) risk, few prior studies have addressed this.

Regular exercise is widely known to improve CVD, raising blood HDL-C and reducing blood triglyceride levels and blood pressure [15,16,17]. Aerobic exercise programs can alter ApoA-I, reducing inflammation and therefore the risk of CAD [18,19,20,21,22]. This provides further evidence that aerobic exercise promotes antioxidative, anti-inflammatory, and HDL-mediated positive effects on vascular endothelial function in MS patients.

Health-improvement exercise programs by the American College of Sports Medicine (ACSM) [23], the European College of Sport Science (ECSS) [24], and the American Heart Association (AHA) [25] emphasize the importance of at least five weekly sessions of endurance exercise, or at least three sessions of high-intensity aerobic exercise with at least two sessions of 8–12 repetitions of resistance exercise. Both endurance and resistance exercise help to reduce body weight, body mass index, waist circumference (WC), blood glucose and insulin levels, and blood pressure, while improving blood lipid variables [26,27]. Combining endurance and resistance exercise, or endurance-strength training and modified resistance exercise, has been reported to enhance these benefits [28,29]. We therefore expect appropriate resistance exercise to help prevent aging-related sarcopenia and MS. Studies comparing the effects of endurance exercise and endurance-resistance exercise on body composition and MS-related variables have reported conflicting results, such as that they have almost the same effects [30,31], that endurance-resistance exercise has greater effects on body composition [28,29], and that endurance-resistance exercise has almost no effects on glucose and fat metabolism [32]. In comparing the effects on the antioxidant function of HDL by static versus aerobic exercise—considering that muscle function exercise can help prevent metabolic diseases—static exercise had little effect, while aerobic exercise effectively improved antioxidant function; it has been reported that this can help reduce blood pressure [18]. However, it has been widely reported that resistance exercise can improve CVD risk factors and inflammatory markers [33], and MS-related risk factors such as diabetes [34,35]. Based on the finding that aerobic and resistance exercise had almost the same effects in diabetic patients, Yang et al. [36] concluded that resistance exercise may achieve this.

Longitudinal associative studies between exercise and HDL dysfunction have reported mixed results [37,38]. Although HDL-C structure and function are known to be associated with MS, and endurance-resistance exercise may help to maximize the anti-inflammatory effects of exercise by altering vascular endothelial function, few prior studies have addressed the effects of the form of exercise in MS. The aim of this study is to confirm the possibility that dynamic-resistance exercise for training with elastic bands in elderly women with MS can bring about positive changes in blood ApoA-I levels, inflammatory markers, and MS risk indicators. We therefore analyzed the structure of dysfunctional HDL-C based on changes in ApoA-I level following dynamic-resistance exercise in elderly women with MS. Dynamic-resistance exercise training qualitatively altered HDL, mostly by altering ApoA-I levels, relieving vascular inflammation, and improving antioxidant function, physical fitness, and MS risk factors. This supports the development of an effective exercise program to prevent and treat MS, and to reduce its prevalence.

## 2. Materials and Methods

### 2.1. Study Participant Criteria

We enrolled 40 elderly women aged ≥ 65 years; all received a detailed explanation about the objectives and procedures of the study and signed informed consent forms. The subjects were randomly selected elderly women who participated voluntarily through a recruitment notice to members of a local senior welfare center, and it is difficult to consider them as perfect general elderly women. Using the NCEP ATP III (National Cholesterol Education Program-Third Adult Treatment Panel) criteria for MS (Table 1) [39,40], subjects who satisfied at least three of the five criteria were diagnosed with MS (MS group, *n* = 19 (after one dropped out due to a musculoskeletal problem), mean age: 71.42 ± 5.87 years). The control group comprised subjects without MS (*n* = 20, mean age: 68.23 ± 2.56 years). The study protocol, including subject selection and exercise intervention methods, was approved by the Institutional Review Board of Keimyung University (Daegu, Korea) (40525-202101-BR-087-03: 31 January 2021).

### 2.2. Exercise Program

The exercise program was comprised of dynamic-resistance exercise using elastic bands, three times weekly, for six months. A red band was used for acclimation to the resistance load. The dynamic-resistance exercise included large-muscle exercises of the upper and lower extremities: seated rows, elbow flexion, archery pulls for the posterior shoulder, hip flexion, hip extension, and long-sitting ankle plantar flexion, as whole-body activities. Initially, the subjects performed 2–3 sets of 10–15 repetitions, with one minute of rest between sets, for a total of 20 min [41]. The warm-up and cool-down exercises (10 min each) were comprised of mostly dynamic stretching. Exercise volume was increased every two months by adjusting the number of repetitions, sets, and the rest time: 11–13 for the first two months, 12–14 for the third and fourth months, and 13–15 for the fifth and sixth months, based on the Borg’s rating of perceived exertion [42]. On-site monitoring was performed to check for hemodynamic response during exercise, and heart rate was measured after each exercise session.

### 2.3. Body Composition and Blood Pressure

Height was measured using a stadiometer (BSM-330, InBody, Seoul, Korea). Body weight was measured using InBody 3.0 (Biospace, Seoul, Korea). The waist–hip ratio was calculated based on waist circumference, measured at the mid-point between the top of the hip bones and the lowest palpable rib. Hip circumference was measured at the widest part of the hip. Blood pressure was measured twice in a two-minute interval using a mercury sphygmomanometer (HICO, Tokyo, Japan).

### 2.4. Lower Skeletal Muscle Function

Health-related fitness was measured using the senior fitness test proposed by Rikli and Jones [43]. In the tandem gait test of walking ability, subjects walked heel-to-toe, without gaps between the steps, along a 3 m straight line, and the time required was recorded. In the figure-of-eight walking test, the time required to move from the starting point to the finish point on a 3 m figure-of-eight line was recorded. To test lower extremity muscle endurance, a 30 s chair stand test was conducted. To test agility, a timed up and go (TUG) test was conducted: this measures the time required to stand up from the chair, walk 3 m, turn around, and return to the chair. To test cardiopulmonary endurance, we used the physical efficiency index (PEI), calculated as [exercise time (in seconds) × 100/sum of heart rate × 2], after checking heart rate three times after the Harvard step test. To test balance, one-leg standing (left and right legs), with eyes open, was performed, and the time standing with the leg raised up to the knee, while balancing with both arms raised, was measured.

### 2.5. Blood Collection and Biochemical Analysis

After at least 12 h of fasting, 5 mL of venous blood was collected from the antecubital vein and centrifuged. Total cholesterol, triglyceride, and LDL-C levels were analyzed on a Hitachi 7150 auto-analyzer (Hitachi, Ltd., Tokyo, Japan). HDL-C was precipitated using a precipitating agent; the HDL-C level in the supernatant was then measured using an enzymatic method [44]. Blood glucose level was measured using the enzymatic hexokinase (HK) method, using a glucose measurement kit (GLU-HK; Asan Pharm., Seoul, South Korea): 3 µL of the sample was reacted with 320 µL of GLU-HKR-1 and 80 µL of GLU-HK R-2, and absorbance was measured at a main wavelength of 340 nm and sub-wavelength of 415 nm. Insulin level was measured using an insulin ELISA kit (Mercodia, Uppsala, Sweden). Homeostatic model assessment-insulin resistance (HOMA-IR), an insulin resistance marker, was calculated as [insulin (µL/mL) × glucose (mmol/L)/22.5], proposed by Mattews et al. [45].

### 2.6. ApoA-I, Pro-Inflammatory, and Anti-Inflammatory Markers

Blood C-reactive protein (CRP) (DCRP00, P183393 kit; R&D Systems, Minneapolis, MO, USA), SOD2 (KA0528 kit; Abnova, Taipei City, Taiwan), HSP70 (ADI-EKS-715, 02201818A kit; Enzo Life Sciences, Seoul, Korea), and ApoA-I (ab100635, KA0528 kit; Elabscience, Beijing, China) levels were analyzed by enzyme-linked immunosorbent assay (ELISA) [46]. TNF-α, IL-6, IL-4, and IL-5 levels were analyzed by Luminex multiplex bead-based assay (PPX-056_MXCE4A4, 191482000).

### 2.7. Statistical Analysis

All data were processed using SPSS 23.0 for Windows. Results are expressed as mean and standard deviation (SD). Differences between groups and time points (before and after exercise training) were analyzed using two-way repeated ANOVA for the main and interaction effects. In the case of significant interaction effects, we used a post-hoc paired *t*-test between time points by group, and a *t*-test between groups by time point. After 6 months of exercise training, Pearson correlation coefficients were calculated to analyze the relationship between changes in blood ApoA1 concentration and metabolic disease-related variables—including body composition, blood pressure, and changes in blood inflammatory markers—and antioxidant variables. Statistical significance was set to *p* < 0.05.

## 3. Results

### 3.1. Body Composition and Blood Pressure

Changes in body composition and blood pressure after 6 months of exercise training are shown in Table 2. After the exercise program, there was no significant overall change in body weight, while waist-hip ratio WHR and blood pressure tended to decrease. The MS group showed a significant decrease (*p* < 0.05) in WHR, WC, and diastolic blood pressure after exercise training, but there were no significant differences in all items before and after exercise training in the control group. Systolic blood pressure in the MS group was significantly higher (*p* < 0.05) than that of the control group, before and after exercise training, although both groups showed a tendency to decrease after exercise training.

### 3.2. Blood Variables

Table 3 shows the changes in metabolic disease-related blood variables after 6 months of exercise training. Namely, blood triglycerides (TG), fasting glucose, and insulin concentrations showed significant decreases in both groups. After 6 months of exercise training, the blood ApoA-1 concentration and ApoA-1/HDL-C ratio were significantly increased (*p* < 0.05) after exercise training in both groups. In the MS group, fasting glucose and insulin concentrations and HOMA-IR decreased significantly (*p* < 0.05) after exercise training, while blood ApoA-1 concentrations and ApoA-1/HDL-C ratio were significantly increased (*p* < 0.05). Before training, blood insulin concentration and HOMA-IR were significantly (*p* < 0.05) higher in the MS group than in the control group; however, after training, there was no significant difference in blood insulin concentration between these groups.

Changes in antioxidant factors and concentrations of inflammatory markers after 6 months of exercise training are shown in Table 4. SOD2 significantly increased (*p* < 0.05) after exercise training in both groups; similarly, IL-4 and IL-5 significantly increased (*p* < 0.05) in both groups. Blood IL-6, CRP, and TNF-a concentrations showed a tendency to decrease after exercise training in both groups, but with no significant difference. Blood HSP70 concentration showed a tendency to increase after exercise training in both groups, and in the MS group, this increase was significant (*p* < 0.05).

### 3.3. Physical Fitness

The changes in physical fitness factors after 6 months of exercise training are shown in Table 5. Both groups showed improvement in all items after exercise training. In particular, after exercise training, the MS group showed a significant (*p* < 0.05) improvement in cardiorespiratory endurance, muscular endurance (30 s chair stand test), balance (TUG test), and walking ability (straight walking test), while the control group showed only significant improvement in muscle endurance (30 s chair stand test) (*p* < 0.05). However, PEI and TUG remained significantly (*p* < 0.05) lower in the MS group, compared to the control group, both before and after exercise training.

### 3.4. Correlation of Blood Variables with APOA-I

The exercise training-induced changes in APOA-I levels (that is, increases in both groups) were significantly positively correlated with the changes in HDL-C (*r* = 0.399, *p* = 0.032) and HSP70 levels (*r* = 0.364, *p* = 0.038), and significantly negatively correlated with changes in triglyceride (*r* = −0.359, *p* = 0.028), CRP (*r* = −0.355, *p* = 0.038), and TNF-α levels (*r* = −0.366, *p* = 0.049).

## 4. Discussion

Sarcopenia and obesity, key risk factors in aging, can drastically increase the prevalence of metabolic syndrome (MS), via negative effects on arteriosclerosis and insulin resistance. Aerobic exercise is typically applied effectively in exercise programs, as a non-pharmacological approach to preventing and treating sarcopenic obesity. Our study, aiming to improve MS in the elderly, follows the growing awareness of the need to include different forms of exercise, including high-intensity interval training (HIIT) or resistance exercise, to prevent aging-related loss of lean body mass and muscle [47]. However, since lean body mass and detailed muscle analyses were not included in this study, it is difficult to determine their effect on the prevention of sarcopenia. Therefore, additional analysis is required on the effects of various high-intensity resistance training, including HIIT, which has recently been considered effective in preventing cardiovascular disease and increasing muscle mass.

We found that dynamic-resistance exercise for training with elastic bands was associated with significant reductions in the WHR and waist circumference, and significant improvements in cardiopulmonary endurance, balance, agility, and walking ability. Endurance exercise has been reported to reduce body weight and body fat. Compared to endurance exercise alone, combined resistance and endurance exercise was more effective for increasing lean body and muscle mass in diabetic patients [48], for reducing body fat and enhancing lower extremity muscle function [49], and for increasing lean body mass and reducing visceral fat [50]. For obese women, Sanal et al. [28] reported that resistance exercise was more effective than endurance exercise in reducing body fat and maintaining lower-extremity lean body mass. Here, we observed significant reductions in WHR and waist circumference and enhanced physical fitness, following dynamic-resistance exercise for training with elastic bands. Considering the increased risk of MS due to muscle mass loss during aging, our findings provide evidence that such exercise could be an effective method for preventing sarcopenic obesity in the elderly. In need of further consideration is that after 6 months of exercise training, the control group showed a significant increase in WC, while the MS group showed a significant decrease. Such a result can be interpreted as a positive effect of dynamic-resistance exercise, but it is difficult to consider its complete effect, considering the MS group showed a significantly high WC pre-intervention, and there were a limited number of subjects to be measured. Therefore, further analysis is required.

Along with obesity, poor blood lipid values are a key risk factor for CVD [51]. Although exercise effectively improves these levels, studies on its effects have typically focused on quantitative changes in body composition and blood lipids [52]. However, the importance of qualitative analysis of HDL-C, including ApoA-I and ApoB-I, has been emphasized for analyzing its specific effects on MS [53,54]. In particular, a detailed qualitative analysis of HDL-C is considered an important diagnostic indicator for CVD and myocardial infarction [55].

TNF-α and CRP levels are elevated in patients with CVD, causing monocytes and macrophages to infiltrate arterial walls, where they accumulate [56]. In this process, native and reconstituted HDL inhibit ApoA-I mutation, as well as inhibiting the leukocyte adhesion molecule expression by endothelial cells that is activated by inflammatory stimulation [57,58]. Accordingly, HDL neutralizes inflammatory markers [59], inhibiting TNF-α expression [50]. Further, low HDL-C levels are associated with high levels of high-sensitivity CRP [60]. Therefore, qualitative changes in HDL-C affect the expression of inflammation, the primary risk factor in MS.

It has been proposed that an exercise program can effectively promote positive qualitative changes in HDL-C and inhibit inflammation. This is based on the hypothesis that increased ApoA-I-mediated HDL binding, arising from increased activation of myeloperoxidase, raises ApoA-I oxidation, and reduces chronic inflammation [61,62]. However, this mechanism has not been sufficiently demonstrated [63]. In particular, it cannot be proven that this mechanism directly reduces lipoprotein levels, which are risk factors of CVD. Following aerobic exercise, ApoA-I levels were elevated in patients with chronic obstructive pulmonary disease [64]. Moreover, in CVD, exercise training directly affected not only traditional cardiovascular risk factors, but also vascular function and structure [65]. Nonetheless, little has been reported on how exercise affects ApoA-I-mediated qualitative changes in HDL [66]. Here, in our MS group, SBP and fasting glucose were noticeably reduced, and ApoA-I and the ApoA-I/HDL-C ratio increased, following six months of active-resistance exercise training. This suggests that exercise will positively alter HDL-C, mostly via blood ApoA-I, in obese and MS patients, providing important evidence toward improving CVD risk factors [67].

Following the exercise program, we observed reductions in the levels of IL-6, CRP, and TNF-α (pro-inflammatory markers), increases in IL-4 and IL-5 (anti-inflammatory markers), and increases in SOD2 and HSP70 (important inhibitors of oxidative stress), providing evidence that vascular inflammation can be prevented through active-resistance exercise. Inflammation caused by obesity and lack of physical activity has been considered a primary cause of type II diabetes and atherosclerosis [68,69], and is associated with increases in pro-inflammatory cytokines IL-6 and TNF-α [70]. Exercise affects the phenotypic switch involved in macrophage polarization, causing a reduction in pro-inflammatory cytokines IL-6 and TNF-α and a related increase in anti-inflammatory cytokines IL-4 and IL-5, and is thus considered effective in inhibiting inflammation [71]. Moreover, the noticeable reductions in blood glucose, insulin, and HOMA-IR, basic indicators of insulin resistance, that we observed following the program, suggest that exercise may increase anti-inflammatory cytokine levels and improve insulin resistance.

Antioxidative function declines with aging. Exercise activates antioxidant factors, and in particular SOD and HSP70; this has been suggested to inhibit arteriosclerosis and insulin resistance by preventing inflammation. Exercise controls oxidative stress, protecting cardiovascular tissue by activating SOD, thereby reducing the myocardial infarct area in the heart [72]. In particular, long-term exercise training can inhibit oxidative stress, by activating skeletal muscle adaptation and mitochondrial biogenesis; even if oxidative stress occurs, the defensive ability can be effectively enhanced by antioxidative enzymes in the mitochondria [73]. High serum HSP70 levels inhibit atherogenesis, reducing the onset of CAD [74]. In elderly rats, aerobic exercise activated SOD and HSP70 expression in muscle tissues [75]. For male rats on a high-fat diet, 12 weeks of running exercise (60 min per session, five sessions per week) was associated with higher SOD1 levels in the aorta and mesenteric artery, relative to those in the non-exercise group [76]. In a human study, it has been reported that exercise training is effective in restoring the function and structure of the heart and blood vessels with the increase in SOD1 levels [77]. Recent reports indicating that resistance training is less effective than aerobic training in activating antioxidative function [78,79] support the use of dynamic-resistance exercise, such as in our study, for restoring antioxidative function. The difference in effective changes in body composition and metabolic disease-related variables according to exercise types (endurance versus resistance) can be attributed to methodical errors, including the inappropriate composition of exercise programs. Therefore, we emphasize the necessity of continuous research for accurate analysis of the effect of the contents of different exercise programs.

Here, following dynamic-resistance exercise for training with elastic bands, our MS group showed positive changes in body composition, physical fitness, and MS-related blood variables, as well as reduced blood pressure, a basic indicator of direct vascular function. This provides evidence that active-resistance exercise to strengthen muscles can prevent MS during aging. Further, our findings show that it repairs dysfunctional HDL-C and reduces intracellular lipid accumulation by increasing vascular endothelial and cardio muscular function.

The improvements in blood ApoA-I levels following dynamic-resistance exercise may be our most important finding: changes in ApoA-I were significantly correlated with changes in body composition, blood lipid levels, and in inflammatory and antioxidative factors. ApoA-I, the most important component of HDL, ensures the function of HDL by interacting with various proteins. Reduced ApoA-I levels, or loss of its lipid-binding ability, are major causes of loss of HDL function, and the resulting dysfunctional HDL cannot form the stable HDL-C structure. CAD is associated with low serum ApoA-I levels [62], and ApoA-I is found in the HDL-C of CAD patients [80]. Among the Apo proteins, ApoA-I is the most potent antioxidant for preventing LDL oxidation and removing oxidized phospholipids from arterial wall cells and LDL [81]. CAD can thus be predicted from the inflammatory and anti-inflammatory actions of HDL [82]. However, since it showed a low overall correlation coefficient with the amount of change in ApoA-I after the exercise program, continuous analysis is needed through a design that further increases the number of subjects.

## 5. Conclusions

For elderly women with MS, dynamic-resistance exercise for training with elastic bands positively altered body composition, blood pressure, physical fitness, MS-related blood variables, and inflammatory markers. In particular, dynamic-resistance exercise caused qualitative changes in HDL, mostly by altering ApoA-I levels. This provides evidence that, for elderly women with MS, dynamic-resistance exercise for training with elastic bands is important in improving vascular inflammation, antioxidative function, MS risk factors, and physical fitness.

## Figures and Tables

**Table 1 healthcare-10-01982-t001:** Five diagnostic criteria for metabolic syndrome (MS) [39].

Waist Circumference	Women: ≥85 cm [40]
Triglyceride	≥150 mg/dL or under medication
HDL-C	Women: <50 mg/dL or under medication
Blood pressure	≥130/85 mmHg or under medication
Fasting blood glucose	≥100 mg/dL or under medication

HDL-C: high-density lipoprotein cholesterol.

**Table 2 healthcare-10-01982-t002:** Changes in body composition and blood pressure after exercise training.

	Control Group		MS Group		Source	F-Value	*p*-Value
	Pre	Post	Pre	Post	Two-Way Repeated ANOVA		
Body weight (kg)	61.38 6.98	61.42 6.79	61.29 6.89	62.39 7.71	Time	0.019	0.891
					Group	0.283	0.601
					T × G	0.017	0.898
WHR	0.89 0.08	0.88 0.06	0.92 0.05	0.89 # 0.04	Time	7.191	0.015
					Group	0.589	0.453
					T × G	0.406	0.532
WC (cm)	86.89 12.21	88.87 6.78	89.19 5.72	86.50 # 5.69	Time	18.756	0.000
					Group	0.667	0.512
					T × G	0.879	0.486
SBP (mmHg)	121.19 17.72	120.24 13.12	137.46 * 6.47	130.01 8.73	Time	1.223	0.328
					Group	10.404	0.005
					T × G	4.215	0.098
DBP (mmHg)	79.21 5.81	79.32 9.18	83.72 8.16	80.18 # 8.61	Time	4.020	0.056
					Group	0.015	0.772
					T × G	7.305	0.022

Values are mean and standard deviation; * *p* < 0.05, comparison between groups in a *t*-test between groups by time point; # *p* < 0.05, between pre and post within each group in paired *t*-test between time points by group; MS: metabolic syndrome; WHR: waist hip ratio; DBP and SBP: diastolic and systolic blood pressure; WC: waist circumference; T: time, G: group.

**Table 3 healthcare-10-01982-t003:** Change of metabolic syndrome markers after exercise training.

	Control Group		MS Group		Source	F-Value	*p*-Value
	Pre	Post	Pre	Post	Two-Way Repeated ANOVA		
TG (mg/dL)	123.45 54.34	112.82 # 34.68	142.81 * 46.67	135.28 * 45.62	Time	2.301	0.042
					Group	2.203	0.046
					T × G	2.432	0.037
HDL-C (mg/dL)	54.25 3.06	56.42 6.89	48.37 3.09	52.81 11.28	Time	1.206	0.252
					Group	0.632	0.465
					T × G	0.012	0.954
Fasting glucose (mg/dL)	91.78 6.67	89.76 9.56	136.37 * 18.09	115.54 *# 25.61	Time	6.212	0.045
					Group	10.602	0.009
					T × G	3.458	0.068
ApoA-I (ng/mL)	4046.16 1470.46	5094.83 # 1819.58	1951.45 * 693.00	3395.32 *# 934.53	Time	0.410	0.538
					Group	0.188	0.675
					T × G	0.191	0.673
ApoA-I/HDL-C ratio	75.39 22.26	92.16 # 25.83	40.73 * 16.99	68.78 # 27.59	Time	25.155	0.000
					Group	9.236	0.007
					T × G	1.592	0.223
Insulin (μU/mL)	7.38 3.61	7.20 3.20	10.39 * 2.90	8.59 # 1.68	Time	3.243	0.088
					Group	13.796	0.000
					T × G	1.599	0.222
HOMA-IR	1.50 0.74	1.55 0.78	4.54 * 0.70	2.35 *# 6.30	Time	2.202	0.155
					Group	17.502	0.001
					T × G	3.535	0.031

Values are mean and standard deviation; * *p* < 0.05, comparison between groups in a *t*-test between groups by time point; # *p* < 0.05, between pre and post within each group in paired *t*-test between time points by group; MS: metabolic syndrome, TG: triglyceride, HDL-C: high-density lipoprotein cholesterol, HOMA-IR: Homeostatic model assessment for insulin resistance; T: time, G: group.

**Table 4 healthcare-10-01982-t004:** Change of anti-oxidants and inflammatory markers after exercise training.

	Control Group		MS Group		Source	F-Value	*p*-Value
	Pre	Post	Pre	Post	Two-Way Repeated ANOVA		
SOD2 (pg/mL)	50827.10 11646.14	52367.78 # 10900.60	50213.45 12651.40	52223.88 # 13070.93	Time	0.600	0.449
					Group	0.006	0.940
					T × G	0.010	0.920
HSP70 (ng/mL)	0.55 0.16	0.69 0.22	0.55 0.22	0.85 # 0.43	Time	11.518	0.003
					Group	0.531	0.475
					T × G	1.795	0.197
IL-4 (pg/mL)	13.93 1.99	14.03 # 2.34	13.20 1.40	15.43 # 0.76	Time	3.726	0.045
					Group	1.323	0.345
					T × G	1.217	0.284
IL-5 (pg/mL)	17.07 8.94	19.86 # 9.52	16.66 11.42	19.28 # 5.50	Time	3.802	0.047
					Group	3.376	0.083
					T × G	0.963	0.339
IL-6 (pg/mL)	14.31 10.95	11.16 3.18	9.96 1.36	9.59 0.85	Time	2.002	0.174
					Group	1.975	0.177
					T × G	1.237	0.281
CRP (mg/dl)	543.72 283.59	447.50 1189.08	674.91 510.69	461.82 563.61	Time	2.616	0.093
					Group	0.090	0.767
					T × G	1.303	0.269
TNF–α (pg/mL)	9.20 3.03	8.36 1.34	8.14 1.47	7.84 0.89	Time	1.261	0.276
					Group	1.550	0.229
					T × G	0.285	0.600

Values are mean and standard deviation; # *p* < 0.05, between pre and post within each group in paired *t*-test between time points by group; SOD2: superoxide dismutase 2, HSP70: 70 kilodalton heat shock protein, CRP: C-reactive protein, IL; interleukin, TNF–α: tumor necrosis factor–α; T: time, G: group.

**Table 5 healthcare-10-01982-t005:** Change of physical fitness after exercise training.

	Control Group		MS Group		Source	F-Value	*p*-Value
	Pre	Post	Pre	Post	Two Way-Repeated ANOVA		
PEI	123.82 13.82	135.24 13.61	98.45 17.91	120.44 * 18.97	Time	5.101	0.031
					Groups	0.958	0.657
					T × G	2.008	0.302
30-s chair stand (time)	17.54 3.21	28.77 # 3.70	15.81 3.72	24.19 # 3.65	Time	81.511	0.000
					Groups	5.566	0.046
					T × G	1.658	0.370
Balance test (Left) (s)	38.02 8.77	42.63 8.10	34.01 4.30	35.17 8.05	Time	1.042	0.322
					Groups	0.546	0.471
					T × G	0.565	0.463
Balance test (Right) (s)	38.86 5.72	53.13 9.45	27.47 4.01	29.96 * 6.59	Time	2.294	0.149
					Groups	3.657	0.074
					T × G	1.132	0.303
TUG (s)	3.07 3.21	4.09 1.42	7.29 * 3.42	6.21 * 1.32	Time	0.194	0.663
					Groups	13.562	0.004
					T × G	2.453	0.246
Straight walking test (s)	7.74 2.19	7.57 1.80	8.93 # 1.52	7.71 * 2.02	Time	2.539	0.134
					Groups	2.468	0.244
					T × G	1.386	0.257
S-type walking test (s)	15.50 1.55	14.04 1.47	15.33 2.21	15.35 2.77	Time	2.659	0.142
					Groups	0.183	0.680
					T × G	2.809	0.132

Values are mean and standard deviation; * *p* < 0.05, comparison between groups in a *t*-test between groups by time point; # *p* < 0.05, between pre and post within each group in paired *t*-test between time points by group; MS: metabolic syndrome; PEI: physical efficiency index; TUG: timed up and go test; T: time, G: group.

## Data Availability

Not applicable.

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
