# Peer review of "Dynamic Resistance Exercise Alters Blood ApoA-I Levels, Inflammatory Markers, and Metabolic Syndrome Markers in Elderly Women"

_healthcare, 2022, doi:10.3390/healthcare10101982_

Round 1

Reviewer 1 Report

Thank you for submitting your valuable manuscript to this journal.

The aim of the present study was to evaluate the effects of six months dynamic resistance exercise on body composition, physical fitness and MS related variables including ApoA-I, antioxidant factors, and some inflammatory markers in elderly women.

This is a very well designed experimental study on two groups of elderly women (with and without MS) with high standard laboratory tests and functional examinations. Materials and methods are very well expressed and the inclusion criteria are clear in each group, nevertheless it would be better to mention the exclusion criteria and the number of participants who excluded from the study before enrolment.

There are some major points in the report of the study that need to be taken into account.

1-      As is apparent from the title of the paper, the primary outcome of the study is to evaluate the effects of dynamic resistance exercises on blood ApoA-1 level, inflammatory and MS markers in elderly women, not only women with MS. The secondary outcome would be the changes in MS or control group. So it is better to present the results of the whole study participants before and after the intervention first, then to mention the results of each group separately and in comparison with each other. Therefore the results of this study have three parts. The main part is for the whole participants before and after the intervention. The second part is for each group before and after the exercises and the third part is the comparison between groups.
Accordingly, the results and related tables should be revised.

2-      The sample size of the whole project is borderline acceptable, but it is small for subgroup analysis.
How do you justify that resistance exercise increased the WC of the control group (although it is not significant, but it's increased by almost 2 cm) yet, significantly decreased the WC of the MS group (almost 2.5 cm)? Is it due to the small sample size of the groups or the measurement errors?

3-      The data of Table 6 is sufficiently stated in the text. I suggest to remove table 6 from the results.

4-      The discussion of the manuscript must be revised completely. In discussion you must express the results of this study and compare them with other studies whether they are confirmed or not. Then explain the similarities or the logic behind the differences in the results.
In the first paragraph of discussion, you stated that your study follows the needs to different forms of exercise including HIIT to prevent loss of lean body mass and muscle mass, but you did not use this type of exercise in your study and even did not measure lean body mass or muscle mass!

5-      Again in the second paragraph of the discussion, there are too much data on body fat, lean body mass and muscle mass in diabetic patients, and visceral fat that are not related to your study objectives. Moreover, at the end of this paragraph, you reported that your findings provide evidence for preventing sarcopenic obesity in the elderly, but the body weight didn't change in your study and you didn't measure muscle mass to evaluate sarcopenia! Line 260, SAA and line 263, Apo C-III protein are not related to your study!

6-      In addition to those gaps in the discussion, there are too many cited references on elderly or diabetic rats (73-77). I’m sure that you can find enough references on human subjects.

7-      The final paragraph of the discussion again contains many lab tests that are not measured in this study and must be removed. (such as lipoprotein-associated phospholipase A2 (Lp-PLA2), paraoxonase1 (PON1), lecithin cholesterol-acyltransferase (LCAT), ApoA-â…¡, and ApoA-â…¥, …)

8-      The conclusion of the study is limited to elderly women with MS, but finally you concluded that this study provides evidence for the whole elderly women!

Some minor points to consider:

1-      Since there is no such thing as passive-resistance exercise, it is better to use dynamic-resistance exercise for training with elastic bands instead of active-resistance exercise.

2-      Line 11, the correct word for MS is metabolic syndrome, not metabolic disorder.

3-      Line 36, prognostic factors for CVD, not diagnostic indicators of CVD!

4-      Line 65, an important reason for the conflicting results is the variability in the definition and use of the types of exercises in different studies. Aerobic, dynamic, endurance, static and resistance exercises are sometimes used inappropriately, causing conflicting results. It would be great if you explain this misconception in the discussion.

5-      Line 69, reference no. 18 is on the effects of aerobic and isometric exercises on dysfunctional HDL in hypertensive patients! That is different from the benefits of exercise in preventing MS. Moreover, it is better to use identical words for the type of exercises that are used in the original paper.

6-      Line 90, please explain the complete word for NCEP and add it to the table 1 legend.

7-      Line 107, exercise volume was increased every two months, not exercise intensity!

8-      Line 111, to check for hemodynamic response during exercise.

9-      Line 115, using a stadiometer.

1-   Please explain the symbols **, ##, and ### in all the tables.

Thank you.

Author Response

Response to Reviewer 1 Comments

We appreciate the detailed comments of the reviewer. We think these points will be of great help in improving the quality of this article. Therefore, our authors have sincerely tried to revise this part. Again, we would like to thank the reviewer for his detailed comments.

Major points

Point-1 :  As is apparent from the title of the paper, the primary outcome of the study is to evaluate the effects of dynamic resistance exercises on blood ApoA-1 level, inflammatory and MS markers in elderly women, not only women with MS. The secondary outcome would be the changes in MS or control group. So it is better to present the results of the whole study participants before and after the intervention first, then to mention the results of each group separately and in comparison with each other. Therefore the results of this study have three parts. The main part is for the whole participants before and after the intervention. The second part is for each group before and after the exercises and the third part is the comparison between groups. Accordingly, the results and related tables should be revised.

Response-1 : According to the comments of the reviewer, all the explanation sentences of the results have been modified as follows.

‘Neither group showed changes in body weight after the training program, whereas the MS group showed a significant reduction in waist hip ratio (WHR) (p < 0.05; Table 2). The control group showed no significant differences in body composition and blood pressure after training, whereas the MS group showed a significant reduction in waist circumference (WC) and diastolic blood pressure (DBP) (p < 0.05; Table 2).

Before training, the MS group showed significantly higher systolic blood pressure (SBP) and fasting glucose than the control group (p < 0.05). Comparing the pre- and post-training values, the MS group showed noticeable reductions in SBP and fasting glucose, and a significant reduction in DBP (p < 0.05) (Tables 2, 3). Both groups showed significant increases in ApoA-1 levels and in the ApoA-I/HDL-C ratio (p < 0.05; Table 3). Before training, the ApoA-I/HDL-C ratio was significantly lower in the MS group than in the control group (p < 0.05); after training, these values were increased in the MS group, showing no significant difference between the two groups. Before training, blood insulin and HOMA-IR were significantly higher in the MS group than in the control group (p < 0.05); after training, these values were noticeably reduced in the MS group, showing no significant difference between the two groups.’

  • Changes in body composition and blood pressure after 6 months of exercise training are shown in Table 2. After the exercise program, there was no significant overall change in body weight, while waist-hip ratio WHR and blood pressure tended to decrease. The MS group showed a significant decrease (p < 0.05) in WHR, WC, and diastolic blood pressure after exercise training, but there was no significant difference in all items before and after exercise training in the control group. Systolic blood pressure in the MS group was significantly higher (p < 0.05) than that of the control group, before and after exercise training, although both groups showed a tendency to decrease after exercise training.

Table 3 shows the changes in metabolic disease-related blood variables after 6 months of exercise training. Namely, blood triglycerides (TG), fasting glucose, and insulin concentrations showed significant decreases in both groups. After 6 months of exercise training, the blood ApoA-1 concentration and ApoA-1/HDL-C ratio were significantly increased (p < 0.05) after exercise training in both groups. In the MS group, fasting glucose and insulin concentrations and HOMA-IR decreased significantly (p < 0.05) after exercise training, while blood ApoA-1 concentrations and ApoA-1/HDL-C ratio were significantly increased (p < 0.05). Before training, blood insulin concentration and HOMA-IR were significantly (p < 0.05) higher in the MS group than in the control group; however, after training, there was no significant difference in blood insulin concentration between these groups.

After training, both groups showed significantly increased SOD2 levels (p < 0.05), and the MS group showed significantly increased HSP70 levels (p < 0.05; Table 4). After training, both groups showed significantly increased IL-4 and IL-5 levels (p < 0.05), and reduced IL-6, CRP, and TNF-α levels (non-significant).

  • Changes in antioxidant factors and concentrations of inflammatory markers after 6 months of exercise training are as shown in Table 4. SOD2 significantly increased (p < 0.05) after exercise training in both groups; similarly, IL-4 and IL-5 significantly increased (p < 0.05) in both groups. Blood IL-6, CRP and TNF-a concentrations showed a tendency to decrease after exercise training in both groups, but with no significant difference. Blood HSP70 concentration showed a tendency to increase after exercise training in both groups, and in the MS group, this increase was significant (p < 0.05).

After training, both groups showed improved fitness (Table 5). In particular, the MS group showed significant improvements in cardiopulmonary endurance, muscle endurance (30 s chair stand test), balance (TUG test), and walking ability (straight walking test) (p < 0.05), and the control group showed significantly improved muscle endurance (30 s chair stand test) (p < 0.05).

  • The changes in physical fitness factors after 6 months of exercise training are shown in Table 5. Both groups showed improvement in all items after exercise training. In particular, after exercise training, the MS group showed a significant (p < 0.05) improvement in cardiorespiratory endurance, muscular endurance (30 s chair stand test), balance (TUG test), and walking ability (straight walking test), while the control group showed only significant improvement in muscle endurance (30 s chair stand test) (p < 0.05). However, PEI and TUG remained significantly (p < 0.05) lower in the MS group, compared to the control group, both before and after exercise training.

Point-2 : The sample size of the whole project is borderline acceptable, but it is small for subgroup analysis.
How do you justify that resistance exercise increased the WC of the control group (although it is not significant, but it's increased by almost 2 cm) yet, significantly decreased the WC of the MS group (almost 2.5 cm)? Is it due to the small sample size of the groups or the measurement errors?

Response-2 : The comments of the reviewer are considered appropriate, and the following sentence has been added to the discussion.

‘In need of further consideration is that after 6 months of exercise training, the control group showed a significant increase in WC, while the MS group showed a significant decrease. Such a result can be interpreted as a positive effect of dynamic resistance exercise, but it is difficult to consider its complete effect, considering the MS group showed a significantly high WC pre-intervention and the limited number of subjects to be measured. Therefore, further analysis is required.’

Point-3 : The data of Table 6 is sufficiently stated in the text. I suggest to remove table 6 from the results.

Response-3 : According to the reviewers' comments, Table-6 was deleted and the sentences in the explanation of the results were modified as follows.

‘The exercise training-induced changes in APOA-I levels (that is, increases in both groups) were significantly positively correlated with the changes in HDL-C and HSP70 levels (p < 0.05), and significantly negatively correlated with changes in triglyceride, CRP, and TNF-α levels (p < 0.05; Table 6).’

-> The exercise training-induced changes in APOA-I levels (that is, increases in both groups) were significantly positively correlated with the changes in HDL-C (r = 0.399, p = 0.032) and HSP70 levels (r = 0.364, p = 0.038), and significantly negatively correlated with changes in triglyceride (r =-0.359, p = 0.028), CRP (r = -0.355, p = 0.038), and TNF-α levels (r = -0.366, p = 0.049).

Point-4 : The discussion of the manuscript must be revised completely. In discussion you must express the results of this study and compare them with other studies whether they are confirmed or not. Then explain the similarities or the logic behind the differences in the results.
In the first paragraph of discussion, you stated that your study follows the needs to different forms of exercise including HIIT to prevent loss of lean body mass and muscle mass, but you did not use this type of exercise in your study and even did not measure lean body mass or muscle mass!

Response-4 : According to the comments of the judges, the sentence in the first paragraph of the discussion was modified as follows.

‘Sarcopenia and obesity, key risk factors in aging, can drastically increase the prevalence of metabolic syndrome (MS), via negative effects on arteriosclerosis and insulin resistance. Aerobic exercise is typically applied effectively in exercise programs, as a non-pharmacological approach to preventing and treating sarcopenic obesity. Our study, aiming to improving MS in the elderly, follows the growing awareness of the need to include different forms of exercise, including high-intensity interval training (HIIT) or resistance exercise, to prevent aging-related loss of lean body mass and muscle [47]. However, since lean body mass and detailed muscle analyses were not included in this study, it is difficult to determine their effect on the prevention of sarcopenia. Therefore, additional analysis is required on the effects of various high-intensity resistance training, including HIIT, which has recently been considered effective in preventing cardiovascular disease and increasing muscle mass.’

Point-5 : Again in the second paragraph of the discussion, there are too much data on body fat, lean body mass and muscle mass in diabetic patients, and visceral fat that are not related to your study objectives. Moreover, at the end of this paragraph, you reported that your findings provide evidence for preventing sarcopenic obesity in the elderly, but the body weight didn't change in your study and you didn't measure muscle mass to evaluate sarcopenia! Line 260, SAA and line 263, Apo C-III protein are not related to your study!

Response-5 : According to the reviewers' comments, the sentence in the second paragraph of the discussion was revised as follows, and content not related to this study was deleted.

‘We found that active-resistance exercise was associated with significant reductions in the WHR and waist circumference, and significant improvements in cardiopulmonary endurance, balance, agility, and walking ability. Endurance exercise has been reported to reduce body weight and body fat. Compared to endurance exercise alone, combined resistance and endurance exercise was more effective for increasing lean body and muscle mass in diabetic patients [48], for reducing body fat and enhancing lower extremity muscle function [49], and for increasing lean body mass and reducing visceral fat [50]. For obese women, Sanal et al. [28] reported that resistance exercise was more effective than endurance exercise in reducing body fat and maintaining lower-extremity lean body mass. Here, we observed significant reductions in WHR and waist circumference and enhanced physical fitness, following resistance exercise. Considering the increased risk of MS due to muscle mass loss during aging, our findings provide evidence that such exercise could be an effective method for preventing sarcopenic obesity in the elderly. In need of further consideration is that after 6 months of exercise training, the control group showed a significant increase in WC, while the MS group showed a significant decrease. Such a result can be interpreted as a positive effect of dynamic resistance exercise, but it is difficult to consider its complete effect, considering the MS group showed a significantly high WC pre-intervention and the limited number of subjects to be measured. Therefore, further analysis is required.’

Point-6 : In addition to those gaps in the discussion, there are too many cited references on elderly or diabetic rats (73-77). I’m sure that you can find enough references on human subjects.

Response-6 : According to the reviewers' comments, reference papers for humans were added and some references to animals were deleted from the discussion.

‘In human study, it has been reported that exercise training is effective in restoring function and structure of the heart and blood vessels with the increase of SOD1 levels [77].’

  1. Schmitt, E. E.; McNair, B. D.; Polson, S. M.; Cook, R. F.; Bruns, D. R. Mechanisms of exercise-induced cardiac remodeling differ between young and aged hearts. Exercise and Sport Sciences Reviews 2022, 50(3), 137-144.

Point-7 : The final paragraph of the discussion again contains many lab tests that are not measured in this study and must be removed. (such as lipoprotein-associated phospholipase A2 (Lp-PLA2), paraoxonase1 (PON1), lecithin cholesterol-acyltransferase (LCAT), ApoA-â…¡, and ApoA-â…¥, …)

Response-7 : According to the reviewers' comments, the discussion part related to items not measured in this study was deleted from the discussion. However, in order to explain the importance of Apo-A1, the explanation of other items was left as it is.

Point-8 : The conclusion of the study is limited to elderly women with MS, but finally you concluded that this study provides evidence for the whole elderly women!

Response-8 : Based on the comments of the reviewer, the conclusion sentence has been modified as follows.

‘This provides evidence that, for elderly women, active-resistance training is important in improving vascular inflammation, antioxidative function, MS risk factors, and physical fitness.’

  • ‘This provides evidence that, for elderly women with MS, dynamic-resistance exercise for training with elastic bands is important in improving vascular inflammation, antioxidative function, MS risk factors, and physical fitness.’

Some minor points to consider:

Point-9 :   Since there is no such thing as passive-resistance exercise, it is better to use dynamic-resistance exercise for training with elastic bands instead of active-resistance exercise.

Response-9 : According to the comments of the reviewers, the entire paper was revised as follows.

‘active-resistance exercise.’ -> ‘dynamic-resistance exercise for training with elastic bands’

Point-10 : Line 11, the correct word for MS is metabolic syndrome, not metabolic disorder.

Response-10 : According to the reviewers' comments, it has been modified as follows.

‘metabolic disorder’ -> ‘metabolic syndrome’

Point-11 : Line 36, prognostic factors for CVD, not diagnostic indicators of CVD!

Point-12 : Line 65, an important reason for the conflicting results is the variability in the definition and use of the types of exercises in different studies. Aerobic, dynamic, endurance, static and resistance exercises are sometimes used inappropriately, causing conflicting results. It would be great if you explain this misconception in the discussion.

Response-12 The following sentence was added to the discussion according to the comments of the judges.

‘The difference in effective changes in body composition and metabolic disease-related variables according to exercise types (endurance versus resistance) can be attributed to methodical errors, including inappropriate composition of exercise programs. Therefore, we emphasize the necessity of continuous research for accurate analysis of the effect of the contents of different exercise programs.’

Point-13 :  Line 69, reference no. 18 is on the effects of aerobic and isometric exercises on dysfunctional HDL in hypertensive patients! That is different from the benefits of exercise in preventing MS. Moreover, it is better to use identical words for the type of exercises that are used in the original paper.

Response-13 : According to the reviewers' comments, it has been modified as follows.

‘In a study of the benefits of muscle-strengthening exercise in preventing MS, when the effects of static and aerobic exercise on HDL function were compared, static exercise had almost no effect [18].’

  • ‘In comparing the effects on the antioxidant function of HDL by static versus aerobic exercise—considering that muscle function exercise can help prevent metabolic diseases—static exercise had little effect, while aerobic exercise effectively improved antioxidant function; it has been reported that this can help reduce blood pressure [18].’

Point-14 : Line 90, please explain the complete word for NCEP and add it to the table 1 legend.

Response-14 : According to the reviewers' comments, it has been modified as follows.

NCEP ATP III -> NCEP ATP III (National Cholesterol Education Program-Third Adult Treatment Panel)

Point-15 : Line 107, exercise volume was increased every two months, not exercise intensity!

Response-15 : According to the reviewers' comments, it has been modified as follows.

Exercise intensity -> Exercise volume

Point-16 : Line 111, to check for hemodynamic response during exercise.

Response-16 : According to the reviewers' comments, it has been modified as follows.

for abnormal cardiac function -> for hemodynamic response during exercise

Point-17 : Line 115, using a stadiometer.

Response-17 : According to the reviewers' comments, it has been modified as follows.

using an extensometer. -> a stadiometer (BSM-330, InBody, Seoul, Korea)

Point-18 : Please explain the symbols **, ##, and ### in all the tables.

Response-18 : According to the opinion of the judges, all tables have been modified and unified as follows.

* p < 0.05, comparison between groups; # p < 0.05, between pre and post in each group;

Reviewer 2 Report

In this manuscript, Ahn et al. describe a study of active-resistance exercise alters blood ApoA-I levels, inflammatory markers, and metabolic syndrome markers in elderly Women. Ahn et al. found that for elderly women with MS, activeresistance training is important in improving vascular inflammation, antioxidative function, MS risk factors, and physical fitnes.

The overall subject is meaningful and worthy of study. Generally, the results obtained it is good. I feel that it is suitable for publication in this journal but, after the authors should accept few revisions of their paper, particularly on the following points:

L164, page4: The correlation analysis methods used later in the manuscript should also be introduced in the statistical analysis.

L166-197, page4-5: The results should be divided into different sections, and each section should have a subtitle.

L166-197, page4-5: The results are all tables. It is suggested to draw some important tables into box charts, so that readers can understand the research results more intuitively.

L166-197, page4-5: In the description of all the table, the meaning of * * and # # representatives is not explained in the manuscript. * *p<0.01?? # # p<0.01??

L184, page4: In the correlation analysis, although p < 0.05, the R value is less than 0.4, so the correlation is weak.

L274-275, page10: Corresponding references should be added after this sentence.

Author Response

Response to Reviewer 2 Comments

We appreciate the detailed comments of the reviewer. We think these points will be of great help in improving the quality of this article. Therefore, our authors have sincerely tried to revise this part. Again, we would like to thank the reviewer for his detailed comments.

Point-1 : L164, page4: The correlation analysis methods used later in the manuscript should also be introduced in the statistical analysis.

Response-1 : The following sentences have been added in statistical analysis  according to the comments of the reviewer. ‘After 6 months of exercise training, Pearson correlation coefficients were calculated to analyze the relationship between changes in blood ApoA1 concentration and metabolic disease-related variables—including body composition, blood pressure, and changes in blood inflammatory markers—and antioxidant variables.’

Point-2 : L166-197, page4-5: The results should be divided into different sections, and each section should have a subtitle.

Response-2 : The following sub-titles have been added according to the comments of the reviewer.

3.1 Body composition and blood pressure

3.2 Blood variables

3.3 Physical fitness

3.4 Correlation of blood variables with APOA-I

Point-3 : L166-197, page4-5: The results are all tables. It is suggested to draw some important tables into box charts, so that readers can understand the research results more intuitively.

Response-3 : Appropriate lines were added according to the judgment of the reviewer.

Point-4 : L166-197, page4-5: In the description of all the table, the meaning of * * and # # representatives is not explained in the manuscript. * *p<0.01?? # # p<0.01??

Response-4 : It has been appropriately modified according to the comments of the reviewer. Based on the reviewers' comments, the following amendments were made. All tables have been modified and unified as follows.

* p < 0.05, comparison between groups; # p < 0.05, between pre and post in each group;

 Point-5 : L184, page4: In the correlation analysis, although p < 0.05, the R value is less than 0.4, so the correlation is weak.

Response-5 : The following sentence have been added in discussion according to the comments of the reviewer.

‘However, since it showed a low overall correlation coefficient with the amount of change in ApoA-I after the exercise program, continuous analysis is needed through a design that further increases the number of subjects.’

 Point-6 : L274-275, page10: Corresponding references should be added after this sentence.

Respone-6 : Following the reviewers' comments, the following references were added.

  1. Busnelli, M.; Manzini, S.; Chiara, M.; Colombo, A.; Fontana, F.; Oleari, R.; Potì, F.;Horner, D.; Bellosta, S.; Chiesa, G. Aortic gene expression profiles show how ApoA-I levels modulate inflammation, lysosomal activity, and sphingolipid metabolism in murine atherosclerosis. 2021, Arterioscler Thromb. Vasc. Biol. 2021, 41(2), 651-667.
  2. Schaefer, E.J.; Santos, R.D,; Asztalos, B.F.; Marked HDL deficiency and premature coronary heart disease. Curr. Opin. Lipidol. 2010, 21, 289–297. doi:10.1097/MOL.0b013e32833c1ef6

Reviewer 3 Report

In the submission, Ahn and Kim, conducted an interesting study on revealing effects of active-resistance exercise. I am not an expert in Physical education. I find it is a well-designed study with solid method and the results/findings seems convincing. My detailed comments are as follows:

In the table captions, there are no explanation of **, ##, etc. Please add some explanations or unify the notations only * and #.

Author Response

Response to Reviewer 3 Comments

We appreciate the detailed comments of the reviewer. We think these points will be of great help in improving the quality of this article. Therefore, our authors have sincerely tried to revise this part. Again, we would like to thank the reviewer for his detailed comments.

Point-1 : In the table captions, there are no explanation of **, ##, etc. Please add some explanations or unify the notations only * and #.

Response-1 : Based on the reviewers' comments, the following amendments were made. All tables have been modified and unified as follows.

* p < 0.05, comparison between groups; # p < 0.05, between pre and post in each group;

Round 2

Reviewer 1 Report

Thank you for the revision of your valuable manuscript.

Many comments have been revised in the manuscript, however there are some other points that should be considered for better presentation.

1-   For the title, I suggest “Effects of dynamic resistance exercise on blood ApoA-I level …” or “Dynamic resistance exercise alters blood ApoA-I level …”

2-      The aim of the study is not clearly stated in the introduction. At the end of the introduction section, please express the main purpose of this study in one sentence. You have to precisely specify the target population of the study. Is it generally old women or only elderly women with metabolic syndrome, or both?

3-      Sample selection is the most important part of the study methods. How did you enroll 40 elderly women? How and where did you find them? Are they representative of their population, and can you generalize the study results to this population?

4-      MS is wrongly stated metabolic disorder in line 48!

5-      Line 36, prognostic factors for CVD, not diagnostic indicators of CVD!

The manuscript format is currently acceptable for publication after these corrections.

Good luck.

Author Response

2nd Response to Reviewer 1 Comments

Thank you for your review.

According to the comments of the reviewers, the following modifications have been made.

Point-1 : For the title, I suggest “Effects of dynamic resistance exercise on blood ApoA-I level …” or “Dynamic resistance exercise alters blood ApoA-I level …”

Response-1 : According to the comments of the reviewers, the following modifications have been made.

‘Dynamic-Resistance Exercise for Training with Elastic Bands alters Blood ApoA-I Levels, Inflammatory Markers, and Metabolic Syndrome Markers in Elderly Women’

  • Dynamic Resistance Exercise alters Blood ApoA-I Levels, Inflammatory Markers, and Metabolic Syndrome Markers in Elderly Women

Point-2 :    The aim of the study is not clearly stated in the introduction. At the end of the introduction section, please express the main purpose of this study in one sentence. You have to precisely specify the target population of the study. Is it generally old women or only elderly women with metabolic syndrome, or both?

Response-2 : Following the comments of the reviewer, the following sentence has been added to the end of the introduction.

‘The aim of this study is to confirm the possibility that dynamic resistance exercise for training with elastic bands in elderly women with MS can bring about positive changes in blood ApoA-I levels, inflammatory markers, and MS risk indicators.’

Point-3 :   Sample selection is the most important part of the study methods. How did you enroll 40 elderly women? How and where did you find them? Are they representative of their population, and can you generalize the study results to this population?

Response-3 : Following the comments of the reviewer, the following sentence has been added to the Methods.

‘The subjects randomly selected elderly women who participated voluntarily through a recruitment notice to members of a local senior welfare center, and it is difficult to consider them as perfect general elderly women.’

Point-4 : MS is wrongly stated metabolic disorder in line 48!

Response-4 : According to the comments of the reviewers, the following modifications have been made.

‘metabolic disorder’ -> metabolic syndrome

Point 5 :  Line 36, prognostic factors for CVD, not diagnostic indicators of CVD!

Response-5 : According to the comments of the reviewers, the following modifications have been made.

‘diagnostic indicators of CVD’ -> ‘prognostic factors of CVD’

Reviewer 2 Report

According to the comments of editors and reviewers, the overall quality of this manuscript has been greatly improved after the author's modification. I feel that it is suitable for publication in this journal but, after the authors should accept few revisions of their paper, particularly on the following points:

The meaning of ** should be explained in tables 2 and 3? For example, * * p<0.01.

It is better to add the statistical method used in the table to the notes of each table.

Author Response

2nd Response to Reviewer 2 Comments

Thank you for your review.

According to the comments of the reviewers, the following modifications have been made.

Point-1 : The meaning of ** should be explained in tables 2 and 3? For example, * * p<0.01.

Response-1 : In all tables, the notation of ** has been changed to *, so there is no need for further explanation.

Point-2 : It is better to add the statistical method used in the table to the notes of each table.

Response-2 : Added 'Two-way repeated ANOVA’ to all tables, and modified it as follows.

 * p < 0.05, comparison between groups; # p < 0.05, between pre and post in each group; -> * p < 0.05, comparison between groups in a t-test between groups by time point; # p < 0.05, between pre and post within each group in paired t-test between time points by group;
